# Temperature effects on carbon storage are controlled by soil stabilisation capacities

Iain P. Hartley [1✉], Tim C. Hill[1], Sarah E. Chadburn [2] & Gustaf Hugelius [3,4]

Physical and chemical stabilisation mechanisms are now known to play a critical role in controlling carbon (C) storage in mineral soils, leading to suggestions that climate warming-induced C losses may be lower than previously predicted. By analysing > 9,000 soil profiles, here we show that, overall, C storage declines strongly with mean annual temperature. However, the reduction in C storage with temperature was more than three times greater in coarse-textured soils, with limited capacities for stabilising organic matter, than in fine-textured soils with greater stabilisation capacities. This pattern was observed independently in cool and warm regions, and after accounting for potentially confounding factors (plant productivity, precipitation, aridity, cation exchange capacity, and pH). The results could not, however, be represented by an established Earth system model (ESM). We conclude that warming will promote substantial soil C losses, but ESMs may not be predicting these losses accurately or which stocks are most vulnerable.

[1] Geography, College of Life and Environmental Sciences, University of Exeter, Exeter EX4 4RJ, UK. [2] Mathematics, College of Mathematics and Physical Sciences, University of Exeter, Exeter EX4 4RJ, UK. [3] Department of Physical Geography, Stockholm University, SE-106 91 Stockholm, Sweden. [4] Bolin Centre for Climate Research, Stockholm University, SE-106 91 Stockholm, Sweden. ✉email: i.hartley@exeter.ac.uk

In cold and wet regions, low soil temperatures and/or anaerobic conditions promote the formation of thick organic horizons and peats, resulting in the storage of ~500 Pg of C[1] and helping drive a latitudinal gradient in soil C stocks[2]. However, for soils or horizons that are a mixture of organic and mineral material (referred to as mineral soils), and which contain more than two-thirds of the global soil C stock[3], the role of climatic variables in controlling C storage has been increasingly questioned[4–6]. It is now recognised that, in the absence of anaerobic conditions, the long-term persistence of soil organic matter (SOM) in mineral soils depends more on its accessibility to decomposers than on its intrinsic decomposability[7,8]. Physicochemical stabilisation mechanisms, including the binding of organic matter with mineral surfaces and physical protection within aggregates, limit SOM availability to microbes and allow organic matter to persist. Because these soil properties are so important in determining C stocks, it has been suggested that the role of temperature in determining C storage may have been overestimated, limiting the potential for a substantial release of C from soils as the climate warms[4–6]. However, the extent to which physicochemical stabilisation mechanisms are themselves temperature-sensitive is poorly understood[9,10]. For example, it has both been argued that sorption (stabilisation) reactions should be more[11] and less[12] sensitive than desorption (destabilisation) reactions, with the relative temperature sensitivities potentially varying with the affinity of organic matter for mineral surfaces[12]. Evidence has been produced that the decomposition of older, and more protected SOM, is more sensitive to temperature than rapidly decomposing SOM, suggesting that destabilisation reactions may indeed be highly temperature sensitive[13–15]. However, understanding remains very limited. Thus, the extent to which temperature controls C storage in mineral soils remains controversial, and it is not known whether the C stored in mineral soils with large capacities to stabilise C will be more or less vulnerable to climate warming than the C stored in soils with more limited stabilisation capacities.

In recent years, important initiatives have started to bring together soil data from across the globe to provide databases that can help answer some of the most pressing environmental questions facing humanity, including the impacts of climate change[16]. For example, the World Soil Information (WoSIS) database published a snapshot of the freely available soil profile data from across six continents[17]. Critically, more than 9300 profiles from non-cropland systems included data on both soil texture and soil C storage by depth (Fig. 1a) offering a major opportunity for improving understanding of how stabilisation mechanisms control the effects of climate on soil C storage. This is because soil texture, in particular clay content, has been shown to be a strong predictor of a soil's C stabilisation capacity[3,18–20], directly determining the potential for chemical stabilisation through the formation of organo-mineral complexes, and indirectly affecting physical protection through its influence on aggregate dynamics[21]. Thus, it is now possible to empirically determine whether the effect of temperature on soil C storage is related to SOM stabilisation capacities.

In this study, by analysing the soil profile data[17], we investigated whether the effect of temperature on soil C storage varies with stabilisation capacities, as indicated by soil textural properties. We hypothesised, because of the evidence that the temperature sensitivity of decomposition is greater for older and more protected SOM pools[13–15], that the effects of temperature on soil C storage would be strongest in fine-textured soils with greater stabilisation capacities. In contrast to our expectations, we observed a much greater reduction in C storage with increasing mean annual temperature in coarse-textured soils, suggesting that unprotected pools are most affected by temperature. The patterns were observed independently for warm and cool regions, across different soil sampling depths, and were maintained after accounting for potentially confounding variables. Our findings emphasise the potential for C to be released from soils in response to climate warming, help identify the soil C stocks that may be most vulnerable, and represent a powerful dataset for the evaluation of Earth systems models.

## Results and discussion

**Effects of temperature on C storage in soils with contrasting stabilisation capacities.** Using a space-for-time approach, we defined the effect of temperature on C storage as the proportional reduction in C storage for each 10 °C increase in mean annual temperature. In this context, a value of 1 indicates no change in C storage with temperature, values less than 1 indicate C stocks increase with temperature and values greater than 1 indicate C stocks decline with temperature, with, for example, a value of 2 indicating that C stocks halve for every 10 °C increase in temperature. C storage in the top 50 cm of mineral soil declined strongly with increasing temperature, declining by a factor of ~1.4 per 10 °C (Fig. 1b). Critically, the nature of this relationship was modified by soil clay content; C storage in fine-textured soils with greater stabilisation capacities was affected much less by temperature than C storage in coarse-textured soils (Fig. 2a and Supplementary Fig. 1; factors of up to 1.9 per 10 °C for coarse-textured soils versus factors below 1.2 per 10 °C for finer-textured soils). We also demonstrate that the lower effect of temperature on C storage in fine-textured soils was retained after accounting for potentially confounding variation in precipitation, aridity (actual minus potential evapotranspiration), plant productivity, soil pH and cation exchange capacity (CEXC) (Fig. 2b). While we focus on the top 50 cm, due to the potential for vertical profiles of soil C to be affected by temperature[22], very similar results were observed for the top 20 cm (Supplementary Fig. 2). In addition, the negative relationship between clay content and the effect of temperature on C storage was observed independently both above and below 15 °C (Fig. 3b, c).

The lower effect of temperature on soil C storage in fine-textured soils with greater stabilisation capacities was unexpected given the evidence of the high-temperature sensitivity associated with the decomposition of more protected SOM pools[13–15]. However, the findings from our global analysis are in agreement with a recent Europe-wide synthesis[23], which, by compiling data from soil physical fractionation studies, demonstrated that mineral-associated C stocks varied less with temperate than freer particulate pools. Therefore, there is growing evidence that the effect of temperature on soil C storage is higher in soils containing a greater proportion of unprotected C.

In the literature, there are apparently contradictory conclusions in terms of how C storage varies across fine-scale climate gradients, in which variation in other factors has been minimised. However, these may potentially be resolved by considering differences in the likely extent of SOM stabilisation. For example, on poorly weathered, relatively coarse-textured, silt loam soils in Alaska, mineral soil C stocks declined strongly with temperature[24]. In contrast, in Hawaiian forests growing on fine-textured soils with high concentrations of Al and Fe oxides, very little change in soil C storage was observed across a gradient of 5 °C in MAT. This was despite the fact that, in these Hawaiian forests, C storage in unprotected pools on the forest floor was found to decline strongly with temperature[5]. We suggest that differences in the extent of physicochemical protection in the Alaskan versus Hawaiian soils may explain the contrasting results. Thus, apparently contradictory findings may be resolvable within a single framework in which the relative effect of temperature on C storage in mineral soils declines

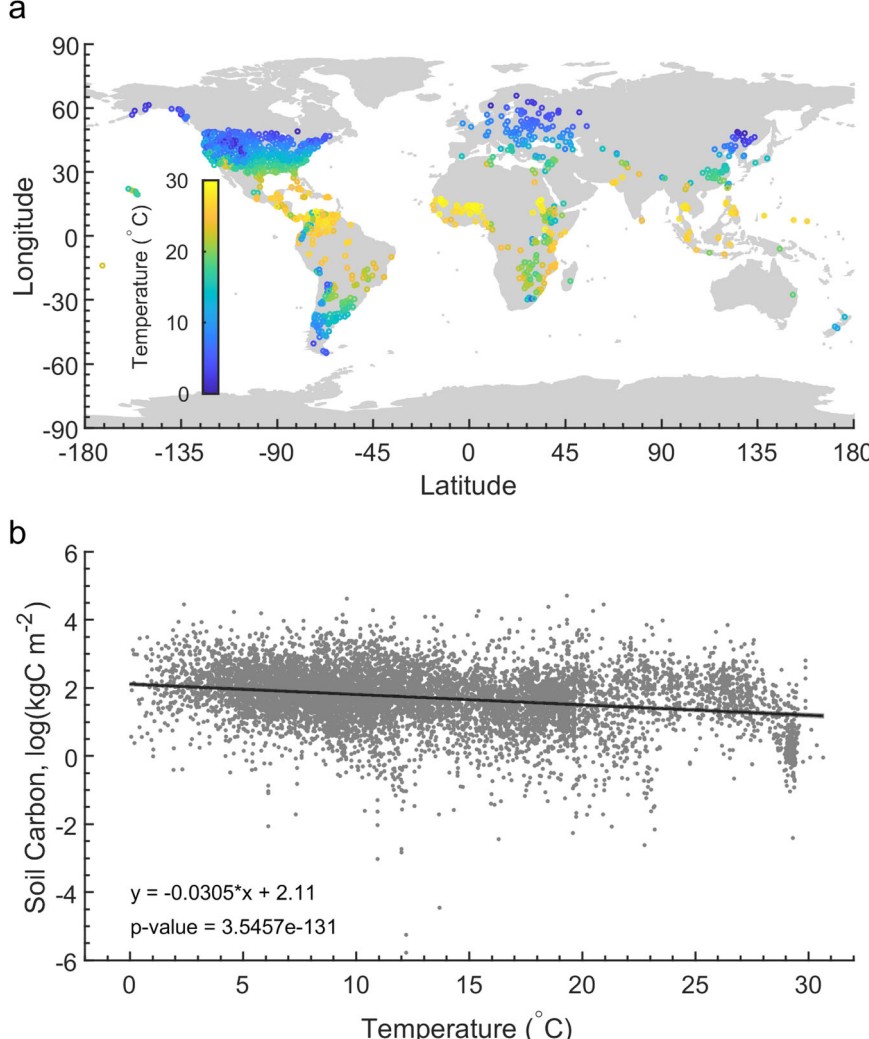

**Fig. 1 Overall effect of temperature on carbon storage.** The location of the soil profiles (**a**) and observed overall relationships between C storage in the top 50 cm of mineral soil and mean annual temperature (**b**). The sampling locations (**a**) are colour-coded by mean annual temperature. The soil C stock data (**b**) have been natural log-transformed and a linear regression fit.

as the soil's physicochemical stabilisation capacity, and the proportion of C in protected pools, increase.

Overall, our analysis identified C stored in high-latitude soils with limited capacities for stabilising organic matter as likely to be most vulnerable to the impacts of climate change. Such stores, therefore, may require particular attention given the high rates of warming taking place in cooler regions. In contrast, the particularly low effect of temperature on C storage in fine-textured soils in warm climates suggests (Fig. 3c) that the C stocks in many tropical soils may be less vulnerable to climate warming. While a soil warming study in a less weathered tropical soil identified the potential for high rates of C release[25], our results are consistent with a recent large-scale analysis that concluded that the temperature sensitivity of soil respiration is generally lowest in tropical environments[26]. However, because C storage in tropical soils has been shown to be potentially vulnerable to drought[27], it should not be concluded that C storage in tropical soils will be unaffected by climate change. Our results do, though, suggest that C stocks in coarse-textured soils at high latitudes are likely to be especially vulnerable to warming (Fig. 3b). Finally, while the dataset contains soil profile information for sites across the full mean annual temperature range investigated (0–30 ºC), and there were data on a minimum of 500 profiles in every 5 ºC

temperature increment, increasing the amount of data available for sites with mean annual temperatures below 5 ºC and greater than 20 ºC would add further confidence to the findings.

Because of their greater stabilisation capacities, fine-textured soils store more soil organic matter[18]. Therefore, fine and coarse-textured soils could contain similar absolute quantities of highly vulnerable C, and the lower effect of temperature in fine-textured soils could reflect the presence of greater quantities of low-vulnerability organic matter[4]. Therefore, it is likely to be very important to quantify stocks of unprotected pools, such as free particulate C, in soils with contrasting stabilisation capacities, and to investigate how such stocks vary with climate[23]. This may make it possible to identify whether there are still important stocks of unprotected organic matter that are vulnerable to climate warming in fine-textured soils with high stabilisation capacities[2].

**Predicting and modelling future rates of C release.** Accurately predicting the response of soil C storage to global warming remains a major challenge. While spatial datasets, such as the ones analysed in this paper, add confidence to the prediction that C will be lost overall and help identify the most vulnerable stocks, they provide limited information on the likely rates or dynamics

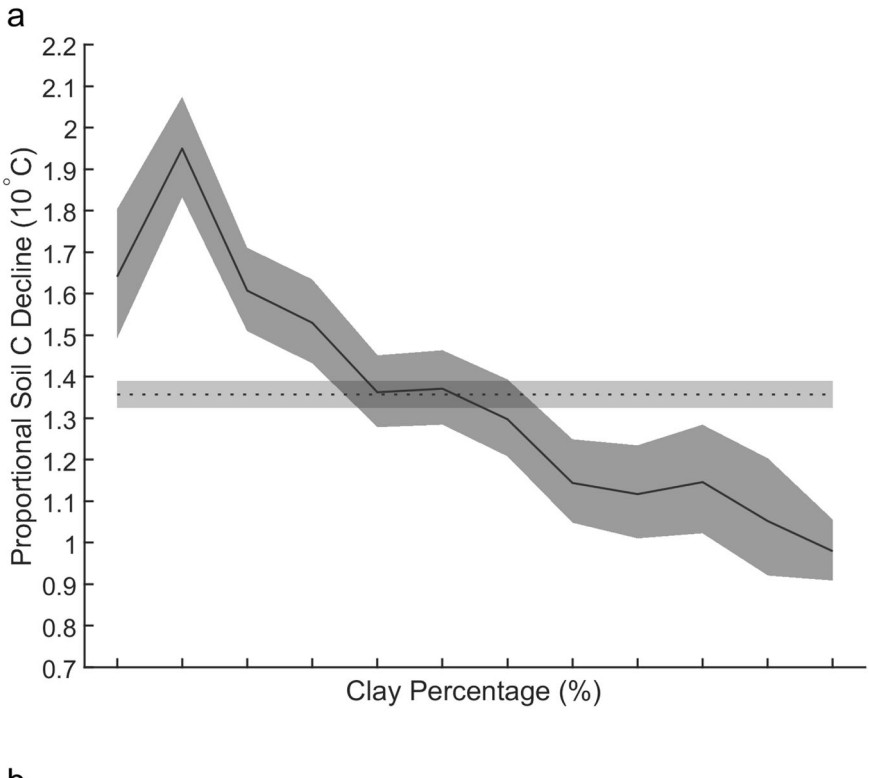

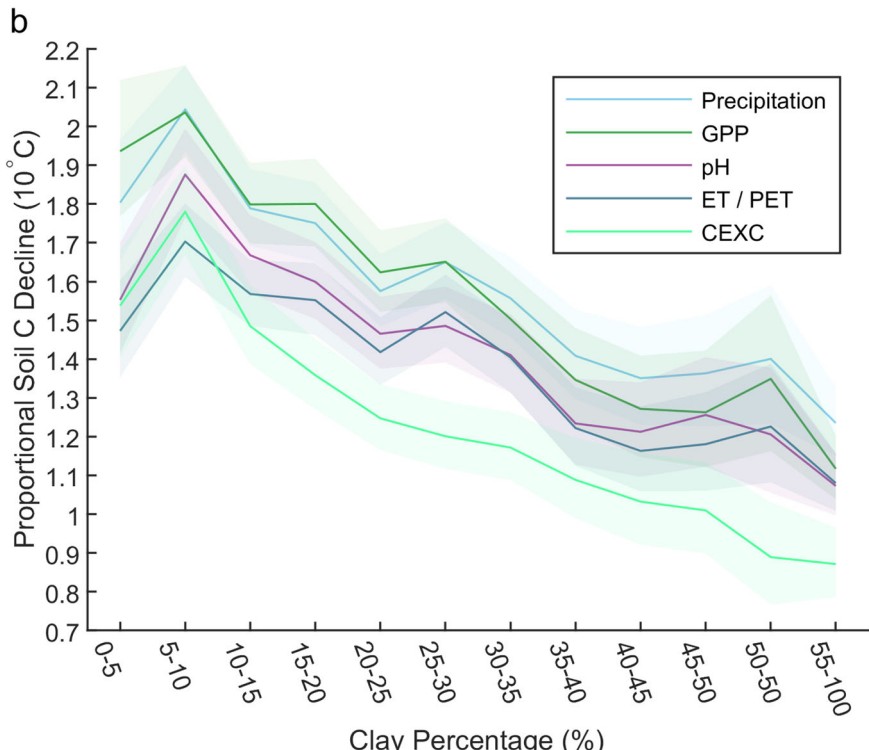

**Fig. 2 Texture effects on temperature–soil carbon storage relationships.** The effect of texture on the relationships between C storage in the top 50 cm of mineral soil and mean annual temperature in the raw data (**a**), and after accounting for potential confounding variables (**b**). The y-axes display the proportional reduction in C storage for each 10 °C increase in mean annual temperature, with higher values indicating greater reductions in soil C with temperature. In panel **a**, the slopes of the relationships (solid line), together with their 95% confidence intervals (dark grey shaded area), are presented for each of the textural categories, with the slope and 95% confidence interval for the full dataset (dotted line and light grey shaded areas) also presented across the graph for comparison. In panel **b**, the relationship between soil C storage and temperature after accounting for variation in annual precipitation (light blue), gross primary productivity (GPP; dark green), soil pH (purple), aridity (ET/PET; evapotranspiration minus potential evapotranspiration; navy blue), and cation exchange capacity (CEXC, light green) are shown. The slopes of these relationships (solid lines) together with their 95% confidence intervals (shaded area) are presented for each of the textural categories.

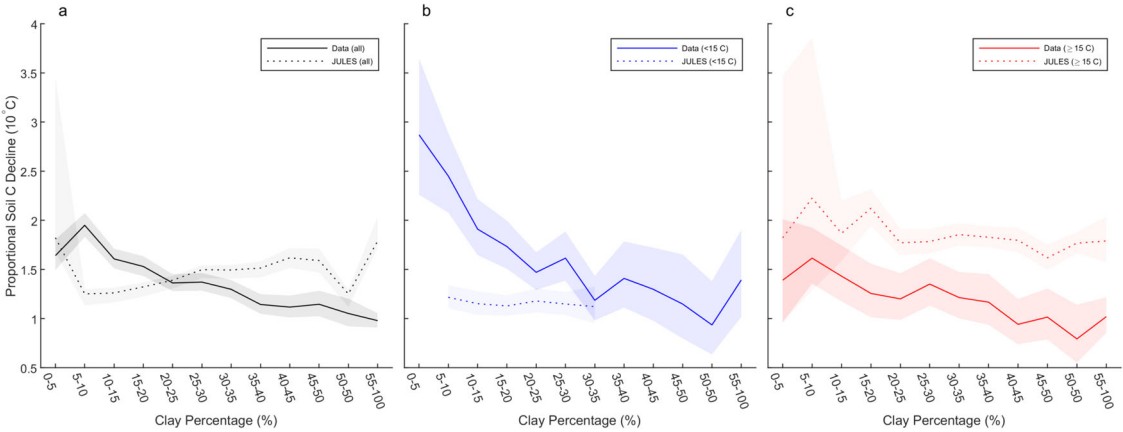

**Fig. 3 Comparison between soil profile data and JULES model output.** The effect of texture on the relationships between C storage in the top 50 cm of mineral soil in the empirical data (solid lines) and JULES output (dashed lines). The slopes of these relationships (solid lines) together with their 95% confidence intervals (shaded area) are presented for each of the textural categories. Results for the full mean annual temperate range (**a**), as well as for subsets of the data for sites with mean annual temperatures below 15 °C (**b**, blue) and above 15 °C (**c**, red) are shown.

of C release. In this context, long-term surveys can be extremely valuable. For example, a recent study in Chinese grasslands was able to detect warming-induced soil C losses since the 1960s and, consistent with the global analysis presented here, coarser-textured soils lost far greater amounts of SOM[28]. Experimental soil warming studies also offer opportunities for further determining the factors controlling soil C storage and predicting rates of C release, although recent syntheses have produced conflicting overall findings[29,30]. Revisiting the networks of warming studies and considering the findings in the context of soil stabilisation capacities and changes in pools of protected and unprotected SOM may allow for a greater understanding of the observed patterns. For example, C losses from subsoils in response to 5 years of whole profile warming were shown to be dominated by the free particulate C pool[31]. Therefore, understanding the responses of different pools to warming may offer the potential to generate mechanistic understanding, even where changes in total C storage have been difficult to identify. It should though be recognised that there are major challenges in accurately quantifying relatively short-term changes in soil C stocks, and there are many other variables that differ between soil warming studies, including contrasting changes in plant productivity and rates of C input driven by interactions between C and nutrient cycling[32]. For these reasons, it may not always be possible to determine the role of soil stabilisation capacities in controlling soil C storage responses to experimental warming[30], and observations collected across space and time will likely remain important for contextualising experimental results.

Soil texture is included as a factor modifying decomposition rates in the terrestrial C cycle modules of a number of Earth systems models (ESMs), including the United Kingdom ESM (UKESM), whose land surface scheme (the Joint UK Land Environment Simulator (JULES)[33]) is based around the Rothamsted C model[34]. Therefore, we investigated whether JULES was already able to represent the patterns that we had observed in the empirical data. In direct contrast to the empirical data, JULES predicted very little variation in soil C with temperature in cooler regions (below 15 °C; Fig. 3b), but predicted a strong effect of temperature on C storage across all textural classes above 15 °C (Fig. 3c). The pattern across the full dataset was confounded by the model simulating only a small number of fine-textured soils at high latitudes (Fig. 3a), and the fact that the relationship between temperature and soil C storage differed so strongly above and below 15 °C. However, crucially,

JULES failed to reproduce the greater effect of temperature on C storage in coarse-textured soils and overestimated the effect of temperature on C storage in fine-textured soils in warmer regions. These findings question whether JULES is identifying accurately which soil C stocks are most vulnerable to the effects of climate warming. This is important given the considerable geographical variation in (1) rates of climate warming and (2) the amounts of C stored in mineral soil horizons. In recent years, there have been major efforts made towards developing models that include physicochemical stabilisation mechanisms and yet can potentially be run at the global scale[35–37]. Testing whether such models can better simulate the observed spatial patterns of C storage in soils with contrasting stabilisation capacities would increase confidence in projections of future changes in soil C stocks[38].

**Limitations and future perspectives.** As well as influencing rates of key biological processes, climatic variables also control pedogenesis, rates of mineral weathering and therefore influence the reactivity of soil surfaces[26,27,39]. Directly determining the binding affinity of mineral surfaces is challenging and would require detailed information on the type of clay minerals present, as well as the abundance of key metal oxides[35,36,40], but there is, currently, insufficient data to assess these more detailed variables at the global scale[35]. However, it has been argued that, at broad spatial scales, soil pH may explain an important proportion of variation in mineral-binding affinities[35,41]. Furthermore, cation exchange capacity (CEXC) varies with the type of clay minerals present and the binding efficiencies of the mineral surfaces[42]. In global analyses, texture, pH and CEXC tend to be the three edaphic factors that correlate most strongly with soil C storage[18,20]. For these reasons, we also accounted for variation in both pH and CEXC, and evaluated whether the relationship between soil texture and the effect of temperature on C storage was retained. We found that it was (Fig. 2b). Thus, we conclude, that within this large dataset, clay content remains a strong predictor of soil stabilisation capacities, both overall, and after accounting for factors that potentially control SOM binding affinities.

While we consider that our analysis of how SOM stabilisation capacities determine the effects of temperature on soil C storage is robust, it is also high level. Thus, there is considerable opportunity to further investigate different vulnerabilities of specific pools of SOM, contrasting the roles of mineral protection versus occlusion in aggregates[7], determining the importance of SOM binding

affinities[40], and linking protection mechanisms with the sources of the organic matter (e.g. microbial versus plant derived[43]). A debate has often revolved around whether climatic versus edaphic factors are more important in controlling patterns of soil C storage. Rather, than focusing on which is more important, for predicting future rates of soil C release, we suggest that a key priority should be on identifying how key edaphic factors determine the vulnerability of contrasting soil C stocks to climate warming. In this context, a recent meta-analysis demonstrated the importance of soil properties in controlling the temperature sensitivity of soil respiration, emphasising how responses to global warming will likely vary substantially between different types of soils in contrasting geoclimatic zones[26].

Using a large global database, we observed declining C storage with temperature in mineral soils, suggesting that there is the potential for strong positive feedback to climate warming. Critically, however, this overall relationship masked differences between soils with contrasting C stabilisation capacities, as indicated by their textural properties. The data suggest that there are stabilised pools of SOM in fine-textured soils that may be relatively insensitive to the impacts of climate change, but that unprotected pools may be substantially more vulnerable to climate warming than currently predicted. Finally, because at least one major ESM was unable to reflect the observed patterns, we argue that ESMs should be evaluated against their ability to simulate the differences in the effects of temperature on C storage in soils with contrasting textural properties in order to reduce uncertainties in projections of the effect of climate change on future soil C storage.

## Methods

**Data sources and processing**. Soil data were obtained from the World Soil Information (WoSIS) database[16] with the 2016 snapshot[17] being downloaded and formatted for analysis in MATLAB (2018a, The MathWorks, Inc., Natick, Massachusetts, United States). About 14,517 profiles contained the necessary information on soil texture, bulk density and C contents, as well as pH and CEXC. This subset of profiles was refined by removing potentially disturbed and managed soil profiles; locations classified as cropland or urban areas in the MODIS IGBP landcover MCD12Q1 dataset[44] were removed from the analysis, leaving a total of 9,326 profiles. Stocks in the top 20 or 50 cm of mineral soil were quantified by first removing surface organic horizons (defined as having gravimetric C contents greater than 20%[45]), and then using data in the next 20 or 50 cm. Profiles with organic horizons below mineral horizons were discounted due to the increased possibility of past disturbance. The stocks were calculated for each sampling layer or horizon based on depth, C content and bulk density. These were then added together, with the depth of the final sampling layer being adjusted to ensure that stocks represented the top 20 or 50 cm of mineral soil. Average clay contents were calculated accounting for variation in bulk density with depth. For each sampling location, long-term average climate information (mean annual temperature and annual precipitation for 1970–2000) was obtained from the WorldClim version 2.0 (http://worldclim.org) database[46]. Soils from sites with a mean annual temperature below 0 ºC were removed, as previous analyses have identified a stronger effect of temperature on C storage in cold climates due to freeze-thaw dynamics[47]. The AppEEARS (https://lpdaac.usgs.gov/tools/appeears/) tool was used to download MODIS data corresponding to the profile locations for the year 2014; annual gross primary productivity (GPP) from the MOD17A3 product[48], 8-day composite evapotranspiration (ET) and potential evapotranspiration (PET) from the MOD16A2 product[49]. Good quality 8-day composite data were used to produce the mean annual ET/PET fraction. Profiles with missing MODIS data were removed from the analysis.

**Data analysis**. Data analysis was completed using MATLAB 2018a. Linear regression was used to determine whether there was an overall relationship between temperature and C storage in mineral soils, with the C stock data being natural log-transformed to reduce the extent to which high C stock outliers drove any relationship and to reflect the hypothesised roughly exponential relationship between decomposition rates and temperature. To determine whether the relationship between soil C storage and the temperature was dependent on soil texture, the different soil profiles were allocated to different soil texture bins based on the % clay content, with each bin representing a 5% increment in percentage clay content (e.g. 0–5, 5–10%). All data above a clay content of 55% was combined into one bin to ensure that the high clay content bins were not characterised by small numbers of observations. The slopes and 95% confidence intervals of the slopes for each relationship were calculated. Differences between texture bins, or between different texture bins and the overall relationship, were considered statistically significant when the confidence intervals did not overlap.

To investigate whether the observed patterns could be caused by confounding environmental variables, linear relationships between mineral soil C stocks (natural log-transformed) and plant productivity (GPP), precipitation (annual) and aridity ET-PET), cation exchange capacity and soil pH (with all calculations using $H^+$ concentrations) were calculated. The same analysis as outlined above was then run on the residuals from these relationships. While all the patterns still held, it is important to note that the aridity calculation already includes temperature in the calculation of potential evapotranspiration which may explain the reduction in the gradient of the slopes (Fig. 2b and Supplementary Fig. 2b).

**UKESM/JULES model runs**. UKESM output data was taken from simulations provided for the sixth coupled model intercomparison project (CMIP6), which represents the state of the art in Earth system modelling[50]. Soil carbon, air temperature (2 m) and clay content data were provided via the Met Office (Andy Wiltshire *pers. comm*). NPP was downloaded from the public data archive (https://data.ceda.ac.uk/badc/cmip6/data/CMIP6/CMIP/MOHC/UKESM1-0-LL, accessed 12/07/2019), where the other variables are also stored. Soil carbon was simulated dynamically by JULES via a Rothamsted-C-type scheme, and NPP was simulated by the dynamic vegetation model in JULES[33]. Clay content was input to the model from an ancillary file. Air temperature was simulated by the coupled atmospheric model in UKESM, with the UKESM CMIP6 configuration being used[51]. Any grid points with annual mean NPP less than $5 \times 10^{-9}$ kg C $m^{-2}$ $s^{-1}$ were filtered out as these covered mostly deserts, which are generally not included or under-sampled in the soil carbon database, and very low NPP leads to very low soil C in JULES that obscures any soil texture effect. Model output was then analysed in the same way as the observations.

## Data availability

All data used in this manuscript are fully open access and available. The soil data were obtained from a published snapshot derived from the World Soil Information database (https://doi.org/10.17027/isric-wdcsoils.20160003), the long-term climate data are available in the WorldClim version 2.0 database (http://worldclim.org), while the MODIS primary productivity, evapotranspiration, and landcover data are available in the MOD17A3 (https://doi.org/10.5067/MODIS/MOD17A3.006), MOD16A2 (https://doi.org/10.5067/MODIS/MOD16A2.006) and MCD12Q1 (https://doi.org/10.5067/MODIS/MCD12Q1.006) databases respectively. The UKESM data from the sixth coupled model intercomparison project (CMIP6) is available in the public data archive (https://data.ceda.ac.uk/badc/cmip6/data/CMIP6/CMIP/MOHC/UKESM1-0-LL).

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

## Acknowledgements

We would like to acknowledge the International Soil Reference and Information Centre (www.isric.org) for compiling and providing access to the WoSIS dataset. S.E.C. acknowledges funding from Natural Environment Research Council (grant no. NE/R015791/1). G.H. acknowledges funding from the Swedish Research Council (2018-04516) and the EU H2020 project Nunataryuk (773421).

## Author contributions

I.P.H. designed the study in discussion with all coauthors. I.P.H. and T.C.H. completed the main analysis of the soils data following an initial analysis by G.H. S.E.C. carried out the analysis of the JULES outputs. The manuscript was drafted by I.P.H. and all authors contributed to the final version.

## Competing interests

The authors declare no competing interests.
