## [Peer Review File · Nature Communications]

Temperature effects on carbon storage are controlled by soil stabilisation capacitiesThe previous round of reviews was completed at another journal

Reviewer Comments:

Reviewer #1 (Remarks to the Author):

The manuscript entitled “Temperature effects on carbon storage are controlled by soil stabilization capacity” explored how the soil organic carbon (SOC) storage relates to temperature and how such relationship is controlled by soil stabilization capacity (as indicated by clay content in this study). The authors found a significant negative relationship between organic carbon stock in mineral soils. Such a relationship is regulated by soil stabilization capacity. Soils with lower clay content (i.e., coarse-textured soils) preserve a steeper gradient of soil organic carbon with temperature, whereas, for soils with higher clay content (i.e., fine-textured soils), a milder gradient was found. The authors compared the findings with the results from one Earth system model (ESM), JULES, and concluded that JULES failed to represent the soil texture’s regulation on SOC storage. Current ESMs may not predict future changes of SOC or its vulnerability accurately. It is critical to understand the underlying mechanisms of SOC stabilization and how the SOC will change under climate change.

The manuscript is easy to read. The message is relatively simple and could be powerful if it pans out. But the authors treated their analysis and arguments too lightly (not rigorously enough) to convince readers of their findings.

RESPONSE 1: We thank the reviewer for recognising the importance of the work and hope that by providing the additional evidence that we have now convinced the reviewer of the robustness of our findings.

First, the term “temperature sensitivity” used in this manuscript is indirectly quantified. The frequently used “temperature sensitivity” describes how soil decomposition (or changes of SOC storage) changes with changing temperature (Davidson & Janssens, 2006). In this manuscript, the authors used spatial patterns of SOC storage over temperature gradients as temperature sensitivity of SOC storage. It is yet not clear how the standing SOC storage is related to its changes under warming (Van Gestel et al., 2018). Therefore, such a gradient may not be considered “temperature sensitivity” in a sense that the community has used (as pointed out by the authors in line 79 - 80 and cited references). Using the derived gradient from this study to infer future SOC storage changes or its vulnerability under global warming (as what the authors discussed in line 27 - 28, line 169 - 175) may not be reliable as the authors acknowledged in line 152 – 153.

RESPONSE 2: Both reviewers requested that we not use the term temperature sensitivity with both considering that this should only apply to temporal rather than spatial variation. We note, however, that the community has used ‘sensitivity’ to refer to changes observed across climatic gradients (e.g. Lugato et al. 2021 Nature Geoscience, looking at POM and MAOM across Europe). Therefore, we now much more explicitly define our use of the term ‘temperature sensitivity’ (lines 78-83 and 119-120). We also hope that by discussing our findings in the context of temporal changes in C stocks from inventories and soil warming studies (lines 166-190) that it is clear exactly where our results contribute to the broader debates on the subject, contextualising the role of spatial gradients versus other measures in evaluating the temperature sensitivity of soil C storage. However, we can still change this terminology if the reviewers remain uncomfortable with the use of ‘sensitivity’.

Second, the authors acknowledged that their conclusion is contradicting a conventional notion that the temperature sensitivity of SOC decomposition is higher for protected SOC than unprotected SOC (line 79 – 82). The authors also proposed that the “apparently contradictory findings may be resolvable within a single framework in which the sensitivity of C storage to temperature is determined by the soil’s capacity for physicochemical

stabilisation” (Lines 146-148). But the authors stop short of explaining the framework or even making hypotheses or speculations. As this contradiction is so critical for readers to understand (and accept) their conclusion, I am not sure how the authors can convince readers of their finding without explaining their framework.

RESPONSE 3: We have now expanded the first section of discussion to more explicitly outline what we meant by the single framework. We now state ‘Thus, apparently contradictory findings may be resolvable within a single framework in which the relative effect of temperature on C storage in mineral soils declines as the soil’s physicochemical stabilisation capacity, and the proportion of C in protected pools, increase.’ (lines 137-142) and provide more evidence to support this framework. We cite the new Lugato et al. 2021 Nature Geoscience study that has identified lower climate sensitivity of mineral-associated C across European ecosystems as well as the new Soong et al 2021 study that identifies losses mainly from unprotected SOM in subsoils following whole-soil warming. While kinetic theory predicts a greater temperature sensitivity of the decomposition of more chemically recalcitrant organic matter, the growing understanding that organic matter persistence in mineral soils is controlled by physio-chemical protection rather than chemical recalcitrance appears to explain our results. To our knowledge, while our study is the first global analysis that has demonstrated greater ‘sensitivity’ of C storage in soils with lower stabilisation capacities, our findings support those of: 1) Lugato et al 2021 based on ~350 locations in Europe, 2) Soong et al 2021 in terms of experimental soil C losses, and 3) Xin et al 2020 (cited in the main paper) in terms of temporal changes in soil C with climate warming. We consider that our work makes a very important contribution to an emerging understanding of which SOM pools are likely most vulnerable to the effects of warming.

Third, The authors used a linear model to retrieve the gradient of SOC storage to temperature and excluding potential confounding effects by other environmental factors (e.g., pH and plant productivity). Did the authors consider multivariate correlations among these variables? Moreover, the authors did not consider the potential collinearity between clay fraction and temperature and its effects on the final conclusion. According to the visual point relationship in Figure 1b and the retrieved values in Figure 3b - c, the gradient of SOC storage to temperature is not consistent across different temperature intervals. The gradient of SOC storage to temperature is larger at cooler zones than warmer zones (Figure 3b - c). At the same time, the clay fraction also exhibits a positive relationship with temperature (see the global clay fraction map at <https://ldas.gsfc.nasa.gov/gldas/soils>). Such patterns raise my concern that whether the observed decreasing “gradient of SOC storage to temperature” with clay fraction is due to the changes of clay or merely a reflection of the collinearity between clay fraction and temperature. The authors may explicitly distinguish the effects of temperature and clay fraction on SOC storage, or use other metrics that can represent soil carbon stabilization capacity to testify whether the main conclusion of this study still holds.

RESPONSE 4: This is a very important issue, and is actually why we carried out our analysis in the way that we did. Although, the relationship between clay content and temperature is weaker in our dataset than in the data the reviewer highlights, an analysis that investigated interactions between clay and temperature could have been confounded in exactly the way the reviewer suggests. This is why we put the dataset into bins of differing clay contents, and as can be seen in Supplementary Figure 1, the data in each bin covers the full temperature range. We can also go one stage further by calculating mean C stock values for each 3°C increment across the temperature range and therefore ensure that the distribution/preponderance of datapoints in any particular section of the temperature range does not affect the fitting of the relationships. Basing the analysis on just 12 datapoints adds a lot of variation and uncertainty to the line fittings. However, the patterns still hold (see Response Figure 1 below, at the end of the response). Thus, we can state with confidence

that it is not the 'collinearity between clay content and temperature' that explains our patterns.

Fourth, in Figure 3b, why JULES only has results in the clay fraction interval 5 – 35? Is that because the input map of clay fraction in JULES present no soils with high clay fraction in cooler regions?

RESPONSE 5: It is not that there are no high clay soils in high latitudes but because JULES is run at a relatively coarse grid scale, which means that the average clay contents in the cooler regions do not exceed 35% in an individual grid cell.

References:

Davidson, E. A., & Janssens, I. A. (2006). Temperature sensitivity of soil carbon decomposition and feedbacks to climate change. *Nature*, 440(7081), 165.
Van Gestel, N., Shi, Z., Van Groenigen, K. J., Osenberg, C. W., Andresen, L. C., Dukes, J. S., . . . Pendall, E. (2018). Predicting soil carbon loss with warming. *Nature*, 554(7693), E4-E5.

Reviewer #2 (Remarks to the Author):

I have reviewed the Manuscript with the title “Temperature effects on carbon storage are controlled by soil stabilization capacities” submitted to *Nature Geoscience*.

The paper deals with an important topic, namely the effects of climate warming on one of the largest terrestrial C stores, soil carbon. The paper is well written and uses a global dataset of soil carbon stocks and a regression approach to identify the dependency of soil C storage on soil texture, climatic variables (MAT) and some other soil variables. I fully agree that this is an important topic and the paper yields some interesting results. The novelty of this work might be a bit limited, but it is a global approach nevertheless, which are rare and important to raise awareness to the variability of soil responses to global change. However, I cannot fully agree with some of the conclusions and approaches taken which I think require some substantial rethinking. A list of my concerns:

RESPONSE 6: We thank the reviewer for their comments and the very constructive way in which they have raised their concerns. We hope that by providing all of the additional analyses that the reviewer has requested that we have been able to address their concerns, and demonstrate that our conclusions are indeed robust.

1. Temperature sensitivity: The paper frequently talks about this but temperature sensitivity is not assessed. The paper uses carbon stocks, not respiration data. There are global respiration database which would actually allow an analyses of sensitivity. But stocks don't really allow that. After all, people wouldn't do temperature sensitive experiments otherwise.

RESPONSE 7: This was also an issue raised by the first reviewer (see also response 2 above). While the community does use the term 'sensitivity' to discuss spatial variation in soil C stocks with climatic variables (e.g. Lugato et al. 2021), it is clear that we did not definite the term precisely enough. Taking on board both reviewer's comments we now provide an explicit definition of what we mean by the term 'sensitivity' (lines 78-83 and 119-120), but can still change this if the reviewer retains their concerns.

2. MAT for temperature. The mean annual temperature variable integrates very very many

things. I doubt it is the best variable to choose to assess temperature sensitivity. Long term higher MAT stimulates also soil formation, weathering, plant growth, nutrient release etc. The 2013 Sistla study in Nature has shown this very well that warming restructures soil C stocks without much net change. The same applies to other slightly warmer and less weathered systems as shown by many studies on areas of glacial retreat etc. The Doetterl study from 2015 in Nature Geoscience for example also showed for larger spatial gradients in Chile (granted, on much less sites than here) that SOC stocks and specific soil respiration follow opposing trends. Higher MAT in many regions might therefore stimulate C input and C stabilization. The point that coarse grained soils are more sensitive to losing C when warming and finer grained ones to a lesser degree is supportive of this argument.

RESPONSE 8: We agree with all the points the reviewer makes here but don't think we argued against any of them in the paper. We take the point regarding MAT, but note that studies that have investigated related issues have also tended to use MAT as the key temperature variable (e.g. Lugato et al. 2021 nature Geoscience). We presented stocks to 50 cm because of the potential for warming to restructure vertical profiles of C storage and the fact that the result is observed in this large volume of soil demonstrates that there is change in C storage with temperature. However, we now also present the results to shallower depths (Supplementary Figure 2, and see also Response 9), and demonstrate the same pattern of declining temperature sensitivity with increasing clay content. We argue that, across this very large dataset, clay content provides a useful measure of soil organic matter stabilisation capacity, but now discuss the role of binding affinities in detail in the discussion (lines 204-229, see also Response 11). In the very important Doetterl 2015 paper, they demonstrate that respiration per unit C stock is lower in the high C soils. This is exactly the pattern that would be expected if greater soil C storage is caused by greater quantities of strongly protected SOC that do not contribute to decomposition rates. Temperature manipulations and respiration studies have suggested that such protected SOC pools may actually be more vulnerable to warming over long time periods, and this was what we expected to find evidence for in the large dataset. However, our results demonstrate that the protected C is less vulnerable than unprotected SOC, and therefore SOC stocks in soils with more limited stabilisation capacities are proportionally more vulnerable.

Our analysis takes variation in GPP and therefore C input into account.

Again, we agree with all the points the reviewer is making and hope that this is now clearer in the restructured discussion section in this new manuscript.

3. The authors integrate the top 50cm of mineral soil. This is quite a large volume for C stocks. C in those depth increment and will show in most ecosystems strong depth patterns. The responses to warming will be highly affected by this since temperature gradients are also strong in this rather large soil volume. In younger and medium aged soils, you might even find strong gradients of weathering, pH etc.

RESPONSE 9: We now also present the patterns in the top 20 cm (Supplementary Figure 2). We focused on 50 cm to avoid biases associated with the vertical redistribution of SOC with temperature that the reviewer highlights above. We hope that showing consistent patterns to different depths enhances the paper, and addresses the reviewer's concerns, together with Response 11.

4. Unclear if the integrated data uses both managed and natural systems or only one of them. Data will be highly skewed towards one or the other in certain environmental systems. This needs to be addressed.

RESPONSE 10: We analysed the dataset with managed systems removed and observed no effect on the patterns, therefore we presented the results for the largest dataset possible. However, the reviewers concerns are very understandable. Therefore, we now present the results with croplands and urban soils removed. Although the total number of profiles has now declined to 9,326, the findings remain the same and a potentially confounding factor is removed.

5. I would have expected more discussion on the weathering and mineralogy effect here given that the authors use some examples to underline their arguments. For example, clays in the tropics are 1:1 dominated, in the temperate zone 1:2 expandable dominated. This seems to be not addressed. A study that identifies clay and texture as the main guiding variable to identify patterns is therefore a bit behind what people are trying to do right now in order to get away from saying: Its all texture only.

RESPONSE 11: We originally had a whole section on clay mineralogy and mineral binding affinities, and clearly should have retained it. We accept all of the arguments that the reviewer raises and have now included further discussion of the limitations of not having more information on clay type, but also we still consider that in this very large dataset, clay content does important information on the potential stabilisation capacity of soils (lines 205-219). We support this with three main arguments:

- 1. In large scale analyses clay content has regularly been shown to be positively related to SOC stabilisation capacities. In the dataset we analysed, there were strong significant positive relationships between soil C stocks and clay content across the whole dataset, and these positive relationships clay and soil C stocks could be observed independently in cool, temperate and warm regions.*
- 2. Studies that have looked at 1:1 versus 2:1 clays have observed slightly reduced C stabilisation capacities per unit clay in 1:1 clays (e.g. Six et al. 2002 Plant Soil **241**, 155-176). Therefore, the trend towards 1:1 clays in warmer regions cannot explain the patterns that we observe as it would have caused a change in the opposite direction to the one we observe (i.e. it could have contributed to declining C storage in fine textured soils at higher temperatures).*
- 3. While we cannot include explicit information of the type of clay, we have now included cation exchange capacity as an additional potentially confounding variable, and still observe the same overall response (Fig. 2b). Because cation exchange capacity should be much lower in 1:1 clays than in 2:1 clays, we are confident that the fact the patterns remain after accounting for cation exchange capacity demonstrates that the patterns are not confounded by changes in clay mineralogy, and clay content remains a strong indicator for soil stabilisation capacity across the whole dataset.*

We also outline the research priorities that need to be addressed to go from this high-level analysis to develop further mechanistic understanding in terms of precisely which types of SOM, in which types of soils, are most vulnerable to climate warming (lines 221-230).

In the light of this points I am also not surprised that the ESM cannot capture the patterns right. Not only because of limitations of the ESM, but also related to the assumptions in this study what MAT and SOC stocks represent. I cannot really recommend to go forward with publication of this study in Nature Geoscience. I don't see that it is really at the forefront of research in the field and does not yield the novelty that I would expect from a publication in NGeo. However, the topic that the authors want to address is really important and maybe should be given another chance of consideration after a larger revision.

RESPONSE 12: We hope that our response has convinced the reviewer of the value of our work. It would be ideal to have detailed information like the Doetterl study across a very wide

range of soils. However, in large-scale analyses the key edaphic factors that can predict SOC have been shown to be clay content, pH and cation exchange capacity (e.g. ref 19 in the manuscript), and it is these that are have used in our analysis. We are confident that by providing all of the extra information that we can convinced the reviewer that our analysis is an important contribution to the subject, and one that can further stimulate efforts to determine how edaphic factors will control the potential responses of soil carbon storage to climate change.

Minor comments:

- Figure 2 Y Axis is not very clear. I interpret it that 1 means no change of soil C with temperature and 2 probably means a lot of change. But this should be clarified.

RESPONSE 13: This has been clarified in the legend for this figure and by defining what we mean by 'sensitivity' throughout the manuscript (lines 78-83 and 119-120).

- I don't think the database should be introduced that it contains about 100000 soil profiles when effectively about 14500 have been used. I would rather give more clarity on the land use and geo-climatic distribution of those points.

RESPONSE 14: We now only refer to the number of profiles we have used.

- I see that the authors here stay away from below 0°C MAT environments. Freeze/Thawing of course happens also in warmer environments so I am not so sure about the rationale for this.

RESPONSE 15: Koven et al. 2017 Nature Climate Change demonstrate a fundamental change in the C storage below a mean-annual temperature (MAT) of 0 °C, that the vertical development of the unfrozen soil profile during the year plays a key role in controlling soil C storage. For this reason, and given the processes we were aiming to investigate, we chose to limit our analysis to locations with a MAT greater than 0 °C and focus on ecosystems with no permafrost present.

Response Fig. 1] The effect of texture on the relationships between C storage in the top 50 cm of mineral soil and temperature. The relationships between temperature and soil C storage in soil with different clay contents (panels a-l represent increasing clay contents in 5% bins, but with all soils with clay contents above 55% combined), and the whole dataset (panel m). Rather than using the full dataset, all the data in 3°C temperature increments has been collapsed down to a single point. Thus every panel has the same number of data points in each part of the temperature range. While this increases variability and uncertainty, it does demonstrate that the patterns remain and thus are not caused by variation in clay content with temperature. In panel n, the slopes of the relationships (solid line), together with their 95% confidence intervals (dark grey shaded area), are presented for each of the textural categories (see panels a-l) together with the slope and 95% confidence interval for the full dataset (dashed line and light grey shaded areas, see panel m).

REVIEWERS' COMMENTS

Reviewer #1 (Remarks to the Author):

The authors have satisfactorily addressed my comments on the previous version of their manuscript. The current manuscript reads very well.

Minor comment:

Line 23, is the word "and" before "after ..." needed here?

Reviewer #2 (Remarks to the Author):

I have reviewed the manuscript submitted by Hartley et al. to Nature communications on the Temperature effects on carbon storage are controlled by soil stabilization capacities. I am reviewer of an earlier version of the manuscript.

I think the authors did a good job in revising the manuscript. I have no bigger concerns now for this paper and I think it is a valuable contribution to the field.

My remaining concerns are quite minor and easy to address:

- I am still not so sure that "temperature sensitivity" is the right word. It is basically the MAT effect or the pattern at least that it holds towards soil carbon storage. I like the formulation in the title (temperature effect). Maybe you can think about using this term more broadly and avoid the use of temperature sensitivity, except for those parts where you talk about potential sensitivities of stabilized and labile C pools in soils.

- In the figures and where applicable in the text when describing relationships I would replace "temperature" by "MAT", since this is what it actually is that we are looking at here.

- l.228: I am not a native speaker, but are those ", " correct here?

- I think 2-3 lines each on the distribution of the included soil datapoints in the main text and the areas which are modeled in JULES is important. (% of datapoints in different climate zones/temperature ranges, how much these zones represent of the total land area that you try to model, etc. For both soil and the NPP constraints that you describe in the model).

- You made some nice points about how spatial patterns of temperature relationships to SOC differed among zones of varying MAT and how this might tie to weathering and soil development history (especially with regards to stabilization mechanisms). Haff et al. 2021 (NCC) and others cited therein recently made a similar argument and tying the two studies together here might be a nice addition to your discussion (since they did use directly measured temperature sensitivity)

REVIEWERS' COMMENTS

Reviewer #1 (Remarks to the Author):

The authors have satisfactorily addressed my comments on the previous version of their manuscript. The current manuscript reads very well.

Minor comment:

Line 23, is the word "and" before "after ..." needed here?

Response: We think that the 'and' helps make it clear that two separate points are being made in this sentence and would prefer to keep it.

Reviewer #2 (Remarks to the Author):

I have reviewed the manuscript submitted by Hartley et al. to Nature communications on the Temperature effects on carbon storage are controlled by soil stabilization capacities. I am reviewer of an earlier version of the manuscript.

I think the authors did a good job in revising the manuscript. I have no bigger concerns now for this paper and I think it is a valuable contribution to the field.

My remaining concerns are quite minor and easy to address:

- I am still not so sure that "temperature sensitivity" is the right word. It is basically the MAT effect or the pattern at least that it holds towards soil carbon storage. I like the formulation in the title (temperature effect). Maybe you can think about using this term more broadly and avoid the use of temperature sensitivity, except for those parts where you talk about potential sensitivities of stabilized and labile C pools in soils.

Response: We have followed the reviewer's suggestion and now refer to temperature effects rather than temperature sensitivity when describing our results throughout the manuscript.

- In the figures and where applicable in the text when describing relationships I would replace "temperature" by "MAT", since this is what it actually is that we are looking at here.

Response: we now state mean annual temperature in the abstract (lines 19-20) and make sure that the temperature effects are clearly defined as being expressed per unit mean annual temperature (line 94-95 and lines 136). The figure legends have also been updated to ensure they refer to mean annual temperature.

- l.228: I am not a native speaker, but are those "," correct here?

Response: We have revised this section to make its structure clearer.

- I think 2-3 lines each on the distribution of the included soil datapoints in the main text and the areas which are modeled in JULES is important. (% of datapoints in different climate zones/temperature ranges, how much these zones represent of the total land area that you try to model, etc. For both soil and the NPP constraints that you describe in the model).

Response: The analysis includes soil profile data and model output across the full 0-30°C temperature range, and no single 5°C increment contains fewer than 500 soil profiles. However, we understand the point the reviewer is making and there are more data available in temperate regions. It would be desirable to increase the proportion of data available from boreal/arctic and subtropical/tropical regions in the future. We now make this point on lines 174-178.

- You made some nice points about how spatial patterns of temperature relationships to SOC differed among zones of varying MAT and how this might tie to weathering and soil development history (especially with regards to stabilization mechanisms). Haff et al. 2021 (NCC) and others cited therein

recently made a similar argument and tying the two studies together here might be a nice addition to your discussion (since they did use directly measured temperature sensitivity)

Response: This is a very useful suggestion. We now include a discussion of the Haaf 2021 NCC study in the section on contrasting responses to temperature in different regions (lines 167-169), and in the section discussing pedogenesis (lines 256-259).